# EEG Emotion Classification Using an Improved SincNet-Based Deep Learning Model

**DOI:** 10.3390/brainsci9110326

**Published:** 2019-11-14

**Authors:** Hong Zeng, Zhenhua Wu, Jiaming Zhang, Chen Yang, Hua Zhang, Guojun Dai, Wanzeng Kong

**Affiliations:** 1School of Computer Science and Technology, Hangzhou Dianzi University, Hanghzhou 310018, China; zenghong519@163.com (H.Z.); wzh@hdu.edu.cn (Z.W.); 182050179@hdu.edu.cn (J.Z.); chenyang87@hdu.edu.cn (C.Y.); zhangh@hdu.edu.cn (H.Z.); daigj@hdu.edu.cn (G.D.); 2Industrial NeuroScience Lab, University of Rome “La Sapienza”, 00161 Rome, Italy

**Keywords:** deep learning (DL), electroencephalogram (EEG), SincNet, SincNet-R, emotion classification

## Abstract

Deep learning (DL) methods have been used increasingly widely, such as in the fields of speech and image recognition. However, how to design an appropriate DL model to accurately and efficiently classify electroencephalogram (EEG) signals is still a challenge, mainly because EEG signals are characterized by significant differences between two different subjects or vary over time within a single subject, non-stability, strong randomness, low signal-to-noise ratio. SincNet is an efficient classifier for speaker recognition, but it has some drawbacks in dealing with EEG signals classification. In this paper, we improve and propose a SincNet-based classifier, SincNet-R, which consists of three convolutional layers, and three deep neural network (DNN) layers. We then make use of SincNet-R to test the classification accuracy and robustness by emotional EEG signals. The comparable results with original SincNet model and other traditional classifiers such as CNN, LSTM and SVM, show that our proposed SincNet-R model has higher classification accuracy and better algorithm robustness.

## 1. Introduction

Emotion is a complicated psychological and physiological state, which originates from the brain’s response to those physiological changes [1], and plays a critical role in our life. Recently, more and more research concentrates on emotion, not only to create affective user interface of human–machine interaction (HMI) applications, but also to evaluate psychological diseases for neural disorder patients, such as Parkinson’s disease (PD) [2], autism spectrum disorder (ASD) [3], schizophrenia [4], depression [5], etc. Therefore, a variety of methods have been employed for emotion recognition, mainly focusing on external behavioral representation such as facial/vocal detection [6,7,8,9,10,11], and internal electrophysiological signal analysis [12,13,14].

Although external behavioral representation sometimes could obtain better emotional classification performance, they are usually subjective [15], and often do not work for neural disorder illness such as autism.

Since the cortical activities directly and objectively reflect changes in emotion states, the measurement of their electrophysiological signals is often used as an analytical signal resource of emotion classification, such as EEG [16,17,18], electrocardiogram (ECG) [19,20], electromyogrphy (EMG) [21,22,23], etc.

Furthermore, in spite of the lower signal-noise-rate (SNR), EEG reveals the characteristics of high time resolution, acceptable spatial resolution, as well as the low-cost and convenient acquisition by portable EEG collection devices, it attracts more and more attention for emotion recognition studies.

The frequency bands of EEG, such as delta (1–3 Hz), theta (4–7 Hz), alpha (8–13 Hz), beta (14–30 Hz), and gamma (31–50 Hz), are still the most popular features in the context of emotion recognition. Moreover, other features such as power spectral density (PSD), event-related de/synchronization (ERD/ERS) and event-related potentials (ERP) perform well in EEG emotion analysis [24,25,26,27]. Recently, various methods are employed for feature extraction and classification in EEG-based emotion recognition. For example, the Fourier Transform-based methods are widely employed for EEG emotion analysis [28,29,30,31,32]. Channel G. et al. [28] combined short-time Fourier Transform (STFT) with Mutual Information (MI) for short-time emotion assessment in a recall paradigm, and acquired better classification accuracy than previous methods. Similarly, Lin et al. [29] also employed STFT, along with asymmetry index (AI) to recognize music-induced emotional responses from brain activity, in addition, the authors further assessed the association between EEG dynamics and music-induced emotion states [30] by fast Fourier Transform (FFT). Similar research could be found in [31,32]. In addition, other analysis approaches, such as wavelet transform (WT) [33,34], common spatial pattern (CSP) [35,36,37], and nonlinear analysis methods [38,39], etc., are also commonly used for EEG emotional characteristics analysis.

To better deal with huge-volume, high-dimensional EEG data, various machine learning algorithms have been used for EEG-based emotion recognition. Soroush et al. [15] employed three well-known machine learning algorithms, including multi-layer perceptron (MLP), K-nearest neighbor (KNN) and support vector machine (SVM) to classify emotion states, and achieved close to 90% recognition accuracy. Zheng et al. [40] took advantage of a discriminative Graphic regularized Extreme Learning Machine (GELM), along with cross-session schemes, to identify stable patterns over time and evaluate the stability of the emotion recognition model. Zhang and Lee [41] adopted an unsupervised method for EEG cluster, and constructed an adaptive neuro fuzzy inference system (ANFIS) to recognize positive and negative emotions. [42] employed SVM and linear discriminant analysis to classify EEG signals into 7 emotion states, and achieved an average accuracy of 74.13% and 66.50%, respectively. Similarly, Mohammadi et al. [43] employed SVM and KNN to detect the emotional states, the experimental results showed the classification accuracy of 86.75% for arousal level and 84.05% for valence level, respectively.

Although those existing machine learning algorithms have found successful applications in the field of emotion recognition, some limitations exist. The limitations include deficient learning inheriting characteristics of training samples [44], and its poor classification accuracy due to the subject dependency of emotion [45]. In addition, the existing machine learning algorithms usually also suffer from severe overfitting [46].

Deep learning (DL) has obtained tremendous improvement since 2015 [47,48], and proven to be much efficient in many complex tasks [46,49,50]. Convolutional neural network (CNN) represents one of the most significant advantages due to its capability of feature extractions directly from raw data without hand-crafted selection, and its success in many challenging classification tasks [51,52]. Recently, CNN-based methods and its variants are applied in the field of EEG emotion recognition as well. Song et al. [16] proposed a Graph CNN (GCNN) to dynamically learn the features and discriminate the relationship between different EEG channels, to improve the EEG emotion recognition. Yang et al. [44] integrated a nonlinear method—recurrence quantification analysis (RQA)—into a novel channel-frequency convolutional neural network (CFCNN) for the EEG-based emotion recognition, acquired an accuracy of 92.24%, and also proved strong relationship between emotional process and gamma frequency band. A hierarchical CNN (HCNN) method was adopted by Li et al. [46] to distinguish the positive, negative and neutral emotion states, the comparable results with stacked auto-encoder (SAE), SVM and KNN shows HCNN yields the highest accuracy.

Although classical CNN methods obtain promising results in dealing with raw data rather than hand-crafted feature extraction, and also express significant potential for practical issues of complex classification, they are still not well be fit for those complex EEG data, due to EEG characteristics of non-stationary, lower SNR, as well as the discrete and discontinuous in the spatial domain [16], especially in the aspect of CNN feature extraction at the first convolution layer.

SincNet [53] is a CNN-based architecture, proposed for speaker recognition by Ravanelli and Benjio in 2018, which is based on parameterized sinc function for band-pass filtering, and only learns those low and high cutoff frequencies from raw data, instead of learning all elements from each filters in traditional CNN architecture. In some extent, though SincNet is designed for speaker recognition, it seems also to be fit for the process of EEG data in those narrow frequency bands of delta, theta, alpha, beta, and gamma. However, in the existing SincNet architecture, the filter banks are specially tuned for speaker recognition applications, and the way the data is fed into the model does not confirm to the characteristics of EEG signals as well. Consequently, in this paper, we propose and improve a SincNet-based architecture, named SincNet-R, for EEG emotion classification and recognition. To better extract EEG features and classify emotion states, we design a group of filters for EEG data, deliberately set the proper length of these filters, at last the way that each sample is fed into SincNet-R is also improved based on the characteristics of EEG data.

## 2. Methods

### 2.1. Preliminary of Sincnet

To better extract the efficient and useful features from the raw data, the first convolutional layer in CNN architecture is quiet crucial for the ultimate classification results, as it not only directly processes those raw input samples, but also is responsible for extracting important detailed classification features from those high-dimensional data such as EEG signals, which can better assistant the subsequent convolutional layers in extracting the features from those complex raw information.

SincNet is such a CNN-based architecture, which could find some meaningful filters through adding some constraints at the first convolutional layer. The architecture of SincNet is shown as Figure 1. In contrast to standard CNN, SincNet has distinct characteristics in signal filtering, adopting a special DNN layer before SoftMax for better classification as well.

In the analysis of EEG signals, compared with the standard CNN, SincNet convolves the input EEG samples with a set of parameterized sinc functions implemented by band-pass filters, the sinc(·) function can be defined as: sinc(x)=sin(x)/x. SincNet convolutional operation at the first convolution layer is denoted as Equation (Equation 1) [53]:(1)$$\begin{array}{cc} Output[n_{num}] & =Input\left[n_{num}\right]\times Fil\left[n_{num}\right]\\ \; & =\sum_{l=0}^{L-1}x\left[l\right]\cdot Fil\left[n_{num}-l\right] \end{array}$$
where Input[·] is the EEG sample chunk inputting into the SincNet model, $n_{num}$ is the number of samples, $Fil[n_{num}]$ is the filter whose length is *L*, Output[·] represents the output after filtering.

In the standard CNN, all the *L* weight parameters of each filter need to be trained in advance for improving classification accuracy, while in SincNet, a predefined function g[·] is used to perform the convolution that only relies on a few learnable parameters θ, the convolution operation of the first convolutional layer of SincNet could also be depicted as Equation (Equation 2) [53]:(2)$$Output\left[n_{num}\right]=Input\left[n_{num}\right]\times g\left[n_{num},\theta \right]$$

The defined function *g* employs a filter bank for band-pass filter in frequency domain, and takes advantage of sinc function to convert to time domain through the inverse Fourier Transform [54], then *g* could be denoted as: $g\left[n_{num},f_{1},f_{2}\right]=2f_{2}sinc\left(2\pi f_{2}n_{num}\right)-2f_{1}sinc\left(2\pi f_{1}n_{num}\right)$, where $f_{1}$ and $f_{2}$ are the low and high cutoff frequency, respectively.

Usually, the parameters of the filters in standard CNN are dynamically modified during its training phase, which will cause some redundancy. While for our proposed model, the *g* function is used for convolution (please see Equation (Equation 2)), which needs to modify only a small number of parameters compared with standard CNNs, and can decrease efficiently parameter redundancy. In addition, the dropout rate in DNN model is set to 0.5. On the one hand, it can improve the model ability of generalization. On the other hand, it will decrease the parameter redundancy, as well as overcome the overfitting for the model.

### 2.2. Design of Sincnet-R

Although better performance has been achieved by SincNet, especially in the aspects of speaker recognition, there are still the following two aspects to be improved when dealing with high-dimensional complex EEG analysis: First, EEG signals have obviously spatial-temporal characteristics. It means, when dealing with EEG signals, it is necessary to analyze the time series features of EEG at each channel, the spatial relationship between different channels should be taken into account as well. But the current SincNet model only processes EEG data with time series features, and maybe loses some spatial ones between different channels, which will lead to miss some EEG spatial details before EEG data is fed into SincNet. Secondly, SincNet filters are designed according to the characteristics of the audio signals in the frequency domain. To some extend, though audio and EEG signals reflect certain characteristics at frequent domain, there also exist some differences: audio signals are mostly very high frequent, while EEG we will analyze is complex, and high-dimensional signals with a frequency between 0.1–70 Hz. Therefore, the preliminary filter design of SincNet is not well fit for feature extraction of EEG signals.

In this paper, we mainly improve SincNet from two aspects to make it more suitable for EEG data analysis. The first is the way that EEG data is fed into the model, due to its spatial-temporal characteristics. In the input data, not only the time series features of EEG data, but also the relationship between spatial channels are included. Secondly, we modify the filter design at the first layer of SincNet, making it better to meet the requirements of temporal and spatial feature extraction of EEG data. On this basis, we propose a SincNet-R model, and apply it to the analysis of emotion EEG data.

Next, we will describe the proposed SincNet-R in detail. EEG is downsampled to 200 Hz, we can then get 200 samples in 1 s. Since a sliding window of 0.5 s is adopted, 100 samples will be acquired, we then take the 100 samples as one epoch, thus a total of 480 epochs in 4 min will be achieved, each containing 100 samples. First, we adopt a sliding window of 0.5 s, to form the initial 62 × 100 EEG samples, in which include 62 channels, and 100 samples of each epoch. Then we can take advantage of reshaping transformation to get a column vector with a dimension of 6200 elements [53]. The EEG data format that inputs into SincNet-R is shown in Figure 2.

In addition, SincNet-R also considers the spatial relationship of EEG data between channels. We randomly take 40% of EEG data as training samples, therefore, the data blocks of 62 × 40 could be acquired for training, as shown in Figure 3.

We then take advantage of Equation (Equation 3) for EEG input data transformation to strengthen the relationship between different sampling channels:(3)$$\begin{array}{cc} I[n] & =(R[n]>>(L\;\mathrm{mod}\;N))|\\ \; & \left(R[n]<<(n-(L\;\mathrm{mod}\;N))\right) \end{array}$$
where *n* is the number of the trained samples, R[n] is the initial data blocks, I[n] is the data blocks after transformation, *L* is the subscript of the data block composed by x and y, x and y are the length of purple and pink data blocks (please see Figure 3), respectively, and the sum of x and y is 62. *N* is the number of EEG sampling channels, here equals to 62.

Concretely, in Figure 3, R[n] is the randomly selective data blocks with the length of 62×40, which consists of 39 data blocks with a length of 62, and another uncertainty data block decided by x and y. Once the trained data blocks is determined, x and y are confirmed as well. Then, according to Equation (Equation 3), we could attach pink data block y above to the top of the purple data block x, the data block thus obtained is the same format as the other 39 ones, and then forms I[n]. Consequently, the trained 62 × 40 data blocks in I[n] are one-by-one correspondence with those 62 channels, and the spatial relationship of these EEG data at different channels is constructed as well.

Figure 4 shows how to segment a testing sample into a few “local” data blocks that could feed these “local” data into SincNet-R. For example, we could easily acquire a 62 × 40 sample as the testing data, then the second “local” data in step size 62, until getting the last “local” data, represented as “first data”, “second data”, and “last data”, respectively. At last, a total of 61 “local” data samples could be acquired for inputting into SincNet-R.

When testing the performance of SincNet-R, we also randomly select 40 out of 100 epochs from each of 62 channels as testing samples. Therefore, we make use of a kind of “localization” operation to construct multiple “local” data with a dimension of 62 × 40 from those 62 × 100 EEG data, as shown in Figure 4. After these “local” data is fed into SincNet-R, the relative probability belonging to a certain class of each “local” data will be acquired by SoftMax in SincNet-R, then the relative probability of all these “local” data in each class is summed, and the class with the largest relative probability sum is the final classification result of the data block.

In addition, we also design a filter bank including 80 filters based on EEG frequency characteristics, as shown in Figure 5. Among them, we divide EEG signal frequency range of 0.1–70 Hz into 76 equal parts to form 76 filters, and the other 4 filters are designed according to the frequency band ranges of theta, alpha, beta and gamma. At last, according to Equations (Equation 1) and (Equation 2), we can get a total of 80 filters that are transformed from the frequency domain to the time domain.

### 2.3. Architecture of Sincnet-R

The architecture of SincNet-R is shown in Figure 6, which is a hybrid model of CNN and DNN. In the section of CNN, it mainly consists of three convolutional layers: C1, C2, and C3. However, in the section of DNN, there are mainly three fully connected (FC) layers: F1, F2, and F3.

The input EEG data into SincNet-R is constructed by a sliding window with 0.5 s time interval. C1 layer uses the sinc function for convolution, and adopts 80 filters to form a filter bank, with a length of 124 elements of each filter. The subsequent two layers, C2 and C3, use the standard convolutional operations, each of which is a filter bank with 60 filters containing 5 elements. In addition, a layer normalization, proposed by Ba J. L. et al. [55], is employed between C1, C2, and C3. The output of C3 layer is directly used as the input of the FC layer F1, which has 2048 neurons. Also, we use a batch normalization strategy for solving the issue of internal covariate shift [56] between FC layers F1, F2 and F3.

In hidden layers, ReLU (Rectified Linear Unit) [57] is used as the activate function, because it is not only more efficient than sigmoid or tanh functions for EEG signal process, but also introduces the sparsity in the hidden layers, and is easy to obtain sparse brain signal feature representations.

For SincNet-R parameter settings, we select RMSProp as the optimizer when SincNet-R is in training, then, we set α as 0.95, and eps as $1\times 10{}^{-8}$. Moreover, two different learning rates are adopted for the training of SincNet-R. The learning rate in the first 200 epochs is set to be 0.001, and 0.0005 after 200 epochs. Moreover, the mini-batch size of each training is 128.In particular, a mel-scale cutoff frequency is employed when initializing C1 convolutional layer.

## 3. Materials

### 3.1. Experimental Protocol

The emotion EEG data of this paper comes from the opening data set “SEED”, recorded by Brain Computer and Machine Intelligence research center (BCMI) of Shanghai Jiaotong University, China. 15 healthy volunteers (Male: Female = 7:8) are recruited as subjects, and are all right-handed. They are required to watch three kinds of emotional film clips in three different stages, those clips include three emotional contents of positive, negative, and natural. There is a total of 15 film clips in one stage, and each clip lasts 4 min. There is 5 s prompt message before each clip is played, 45 s for self-assessment after watching the clip, and then 5 s for resting, the experimental schematic is shown in Figure 7. In addition, a questionnaire will be filled out after the volunteers finish each stage in the experiment.

### 3.2. EEG Recording

EEG is sampled with a frequency of 1000 Hz, and impedance of below 5 KΩ. For data dimension reduction, EEG is then downsampled to 200 Hz. 62-channel EEG signals are collected while the volunteers are watching film clips, and the placement of electrodes conforms to the 10–20 international standard system. To ensure stability and reliability of the experiment, EEG is recorded in three stages, the time interval between two stages is one week or longer. Please visit the website http://bcmi.sjtu.edu.cn/∼seed/index.html, and refer to the publication [58] for more details.

### 3.3. The Establishment of EEG Data Sets

All the electrodes are referenced to the left earlobe. The EEG recording is filtered between 0.1–50 Hz with a band-pass filter [58]. Electrooculogram (EOG) signals are simultaneously recorded, together with independent component analysis (ICA) [57] for the removal of eye-movement artifacts.

In this paper, we construct two categories of data sets for SincNet-R performance testing.

The first is the data set of each subject at each stage, we call them as EEGData_1[i][j][k], which means EEG recorded when the i_th subject watches j_th clip at k_th stage, where i≤15,j≤15, and k≤3. Since each subject will watch 15 clips at each stage of the experiment, and the clips with positive, negative and natural account for one third, respectively, we select 185 samples when each subject watches one clip, and 2775 samples will be acquired for each subject during one stage, of which includes 925 samples of positive, negative, and natural, respectively. Therefore, we get a total of 41,625 (2775 × 15) samples.

The second is the data set for each subject at 3 different stages, we also called them as EEGData_2[i], which means the collected EEG data from the i_th subject during the whole experiment. We then establish the data sets for each subject throughout the experiment. Among them, 1350 samples are selected at each stage, and the total number of samples selected by each subject in 3 different stages is 20,250. A total of 303,750 samples is available for 15 subjects.

## 4. Results and Discussion

In this section, we will discuss the performance between SincNet-R and SincNet, as well as other traditional classification models, including CNN, LSTM and SVM, in terms of accuracy, convergence and robustness of emotional EEG classification.

### 4.1. SincNet-R vs. SincNet

The experiment is run on a GPU with a 1080Ti graphics card under Linux. When our training data reaches about 400 epochs, both SincNet-R and SincNet begin to be convergent and stable, which take about 10 h.

We first test the classification accuracy of SincNet-R and SincNet on EEGData_1, as shown in Figure 8. EEGData_1 is tested just from the perspective of inter-subject. From Figure 8, we can find that after converging, the accuracy rate of SincNet fluctuates around 80%, while that of SincNet-R can always be kept above 90%, and the highest accuracy can even reach 95.7%. It means that SincNet-R has better classification accuracy rate than SincNet.

The results in Figure 8 show that although there are significant differences in EEG signals between different subjects, both SincNet-R and SincNet have higher classification accuracy, especially SincNet-R, which can learn brain activities by EEG better and more stably, and provide excellent classification results for the subjects.

In addition, we also test the performance of both models on EEGData_2 from the perspective of intra-subject, as shown in Figure 9. In the emotional classification task for inter-subject, the accuracy of SincNet-R is higher than that of SincNet, which is at least above 90%, while SincNet’s classification accuracy rate is between 75% and 85%.

In general, the classification accuracy of intra-subject should be higher than that of inter-subject, but in our experiment, it is not that case, because we mainly consider the classification performance of intra-subject EEG at different stages. As we know, EEG signals will exhibit significant differences over time, while EEGData_2 is collected in three stages, and each stage is collected at a time interval of at least one week. Therefore, the lower accuracy rate on EEGData_2 than that on EEGData_1 is mainly due to the differences in EEG signals over time.

### 4.2. Compared to Other Classical Models

To further validate the performance of our proposed SincNet-R on EEG classification, we also compare the classification performance of SincNet-R with other classifiers including SincNet, SVM, LSTM and standard one-dimensional CNN on EEGData_1 data set, respectively, moreover, 10-fold cross-validation is performed to verify the accuracy and generalization capabilities of our proposed SincNet-R.

Among them, please refer to Section 2.3 for the parameter settings of SincNet-R, and the publication [53] for those of SincNet. The parameter settings for the other three classifiers are as follows: CNN that we select is a standard one-dimensional convolutional model, which consists of three layers, the number and length of the filters for each layer are 80, 60, 60 and 124, 5, 5, respectively. In LSTM, similar to SincNet-R, RMSProp is also chose as the optimizer. We then set the number of hidden neurons in each LSTM unit as 512, timestep as 100, data_input as 62, and learning rate as 0.001. While in SVM, RBF (Radial Basis Function) is adopted as the kernel function, and OVO (one-versus-one) mode as the multi-classification decision mechanism.

The classification results are shown in Figure 10, which are the average values after the models converge. After cross-validation, SincNet-R has a much higher accuracy than the other four classifiers, and the highest accuracy reaches 95.22%, please see Table 1 for detailed information on the average classification accuracy of these five classifiers. The results indicate the improved SincNet-R could better classify emotion EEG signals.

### 4.3. Variance and Convergency Analysis of Sincnet-R

To verify the robustness of SincNet-R, we also conduct the variance analysis of the five models (please see Table 2), as well as the convergency by loss rate of both SincNet-R and SincNet, as shown in Figure 11.

From Table 2, we could find SincNet-R, which is similar to LSTM, also has better performance of robustness than the other three classifiers in the field of emotional EEG identification. The main reason is that SincNet-R has designed filter banks specifically for EEG signals, making EEG feature extraction more stable.

Moreover, we also compare the convergence of the two models SincNet-R and SincNet. Figure 11 depicts the loss decrease of SincNet-R and SincNet during training. The loss rate of SincNet fluctuates greatly and its accuracy has been fluctuating around 80% after reaching 50 epochs, at which point the loss rate has not converged. While SincNet-R converges quickly, SincNet-R achieves better convergence performance than SincNet after 150 epochs. In other words, when SincNet-R is trained, the loss value is always smaller than SincNet, and it takes less time to drop and keep stable.

The main reason is that SincNet-R improves the first layer of model convolutional layer filter, making SincNet-R better able to meet the requirement of EEG feature extraction. In addition, we adjust the length of the filter bank to 124, which makes it better to maintain its spatial correlation of EEG signals between channels, and fit for the process of 62-channel EEG signals. The experimental results also show that SincNet-R converges faster during training.

## 5. Conclusions

In this paper, we improve and propose a SincNet-based classifier, SincNet-R, which consists of three convolutional layers, and three DNN layers. We make use of SincNet-R to conduct a tri-classification of emotional EEG signals. Furthermore, the classification accuracy between SincNet-R, SincNet, and the other three classifiers, CNN, LSTM and SVM, is compared, the convergence and robustness of SincNet-R are analyzed as well.

The experimental results show that our proposed SincNet-R classifier has better classification performance than other traditional classifiers, and also SincNet-R could converge faster than SincNet when performing the feature extraction of 62-channel EEG signals.

However, our proposed SincNet-R classifier is just verified at SEED data set, and only conduct a tri-classification of EEG signals. In our future work, we will make use of different EEG data sets to validate the proposed SincNet-R classifier, based on that, we will try to conduct multi-label classification work on EEG signals.

## Figures and Tables

**Figure 1 brainsci-09-00326-f001:**
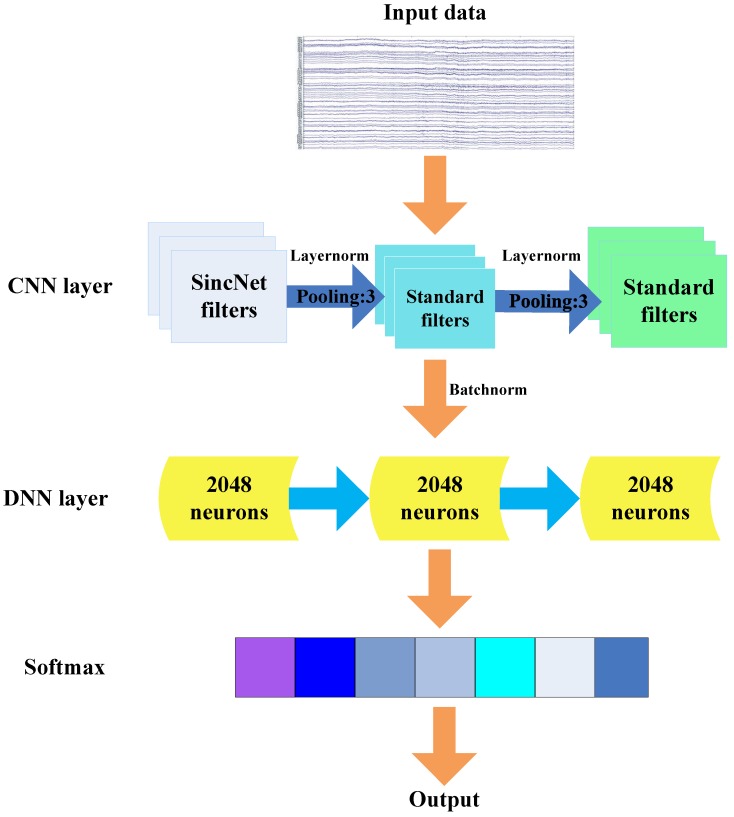
The architecture of SincNet.

**Figure 2 brainsci-09-00326-f002:**
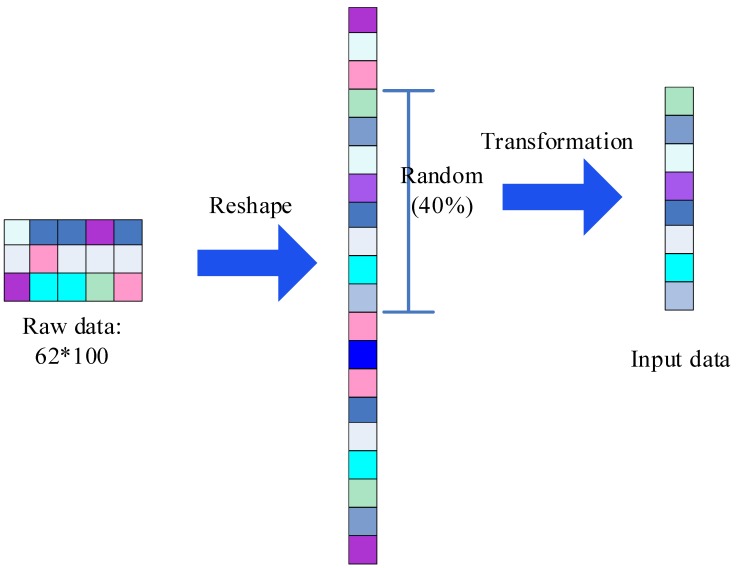
The input EEG data format.

**Figure 3 brainsci-09-00326-f003:**
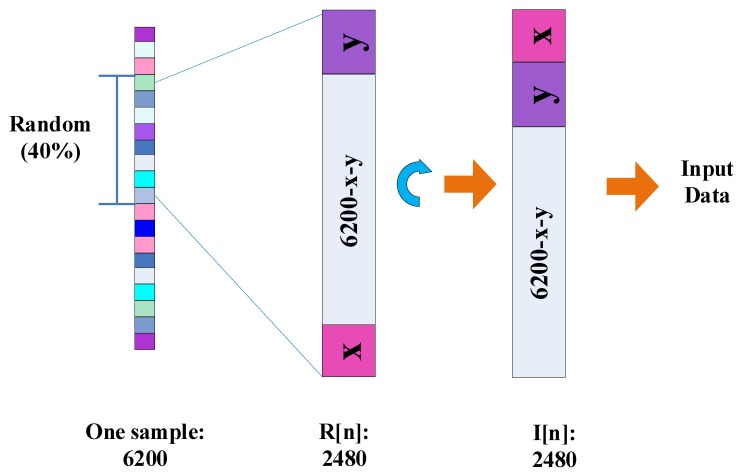
The spatial transformation of the input EEG data.

**Figure 4 brainsci-09-00326-f004:**
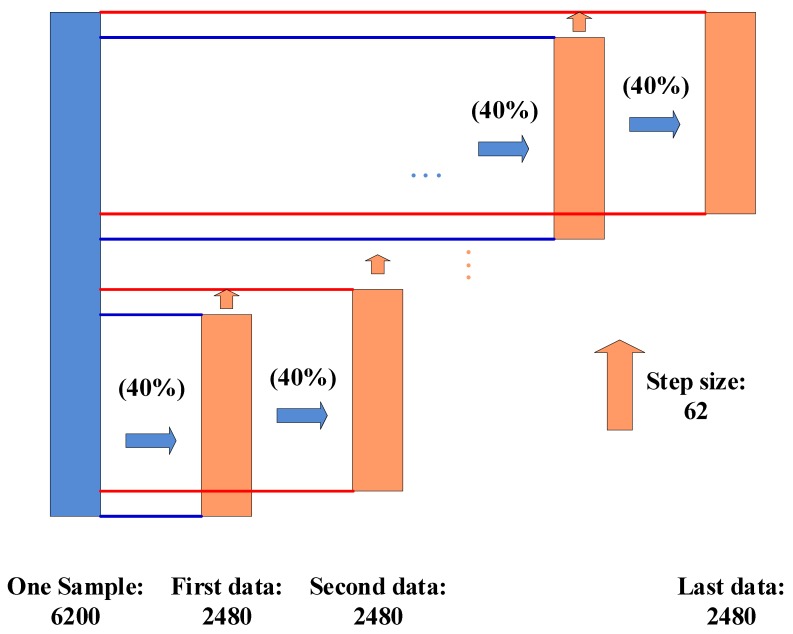
The local data acquired from the sampling.

**Figure 5 brainsci-09-00326-f005:**
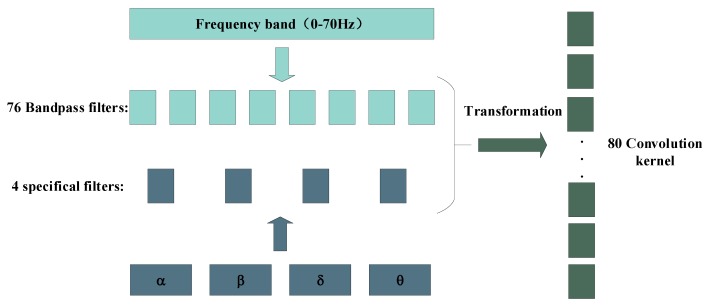
EEG filter bank design in frequency domain.

**Figure 6 brainsci-09-00326-f006:**
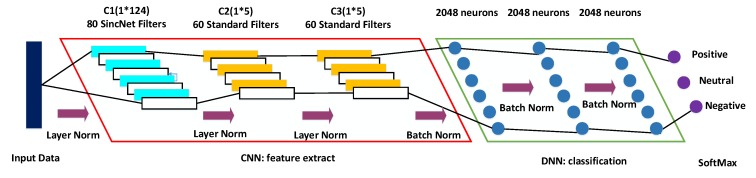
The architecture of SincNet-R for emotion classification.

**Figure 7 brainsci-09-00326-f007:**
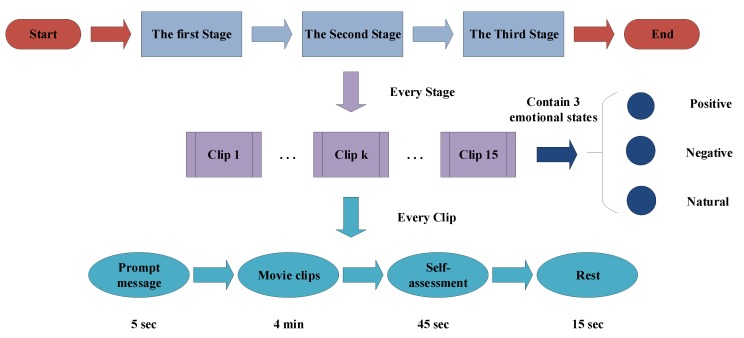
The experimental schematic.

**Figure 8 brainsci-09-00326-f008:**
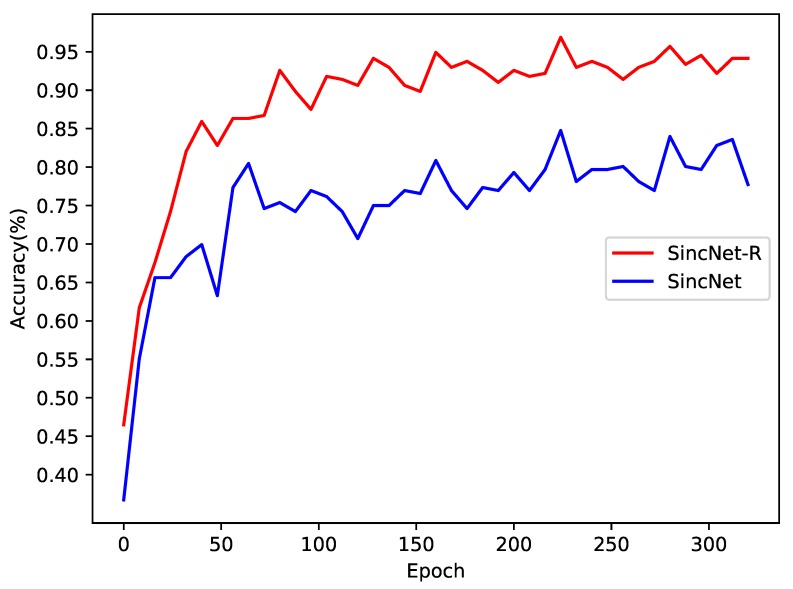
Emotion EEG classification accuracy comparison between SincNet-R and SincNet on EEGData_1.

**Figure 9 brainsci-09-00326-f009:**
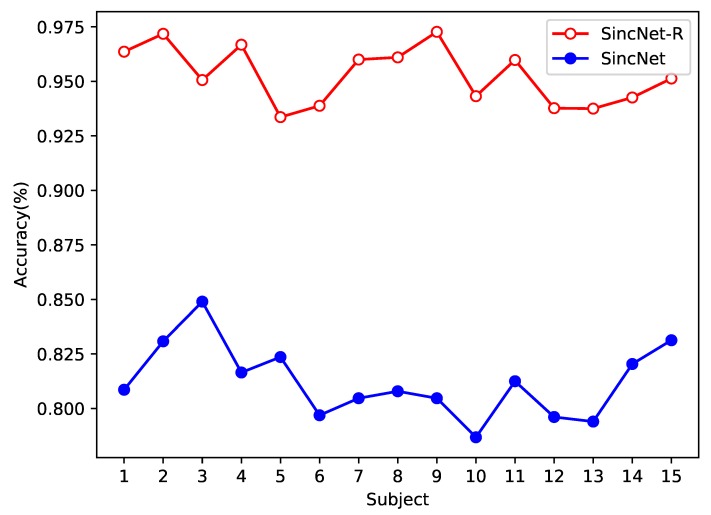
Emotion EEG classification accuracy comparison between SincNet-R and SincNet on EEGData_2.

**Figure 10 brainsci-09-00326-f010:**
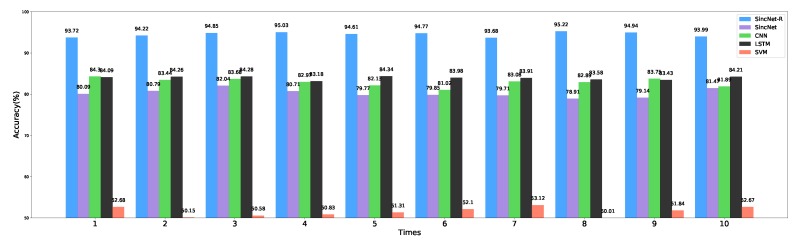
Classification accuracy comparison with SincNet, CNN, LSTM, and SVM.

**Figure 11 brainsci-09-00326-f011:**
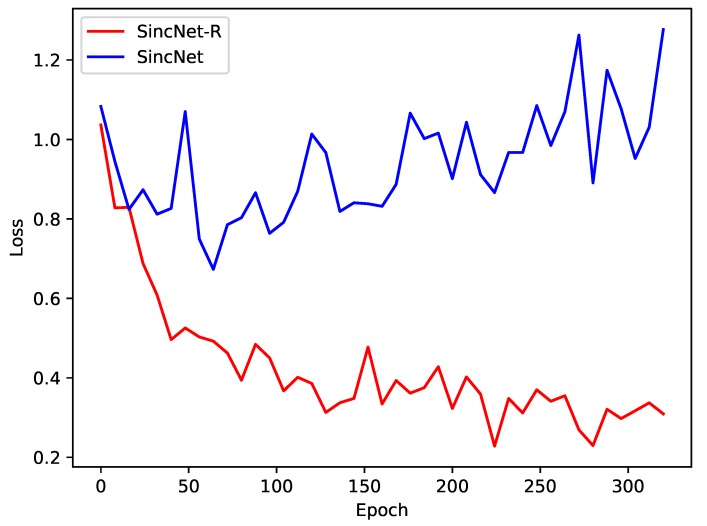
Loss of SincNet-R and SincNet on EEGData_1 and EEGData_2.

**Table 1 brainsci-09-00326-t001:** Average classification accuracy of SincNet-R, SincNet, CNN, LSTM, and SVM.

Model	SincNet-R	SincNet	CNN	LSTM	SVM
Average accuracy (%)	94.503	80.248	82.915	83.926	51.529

**Table 2 brainsci-09-00326-t002:** Variance analysis of SincNet-R, SincNet, CNN, LSTM, and SVM.

Model	SincNet-R	SincNet	CNN	LSTM	SVM
Variance	0.282	0.893	0.877	0.144	1.123

## Data Availability

The emotional EEG data used to support the findings of this study are included in the web site: http://bcmi.sjtu.edu.cn/∼ seed/index.html.

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
