# Peer review of "EEG Emotion Classification Using an Improved SincNet-Based Deep Learning Model"

_brainsci, 2019, doi:10.3390/brainsci9110326_

Round 1

Reviewer 1 Report

Powerline noise in biological recordings (raw EEGs) centered at 50 or 60Hz must be filtered out before biological signals can be of any use. The analyzed EEGs were selected between 0.1-70Hz. Were those EEGs free of the powerline noise? According to Fig. 7, 15 subjects watched 15 clips of movies during EEG recording. It seems that each movie clip was continuous without interruption and differed from the conventional EEG epoch (e.g., a face picture onset followed by a bottom press). On p. 5, it is unclear where these 100 epochs came from. In the CCN and DNN architecture, there are many parameters to be estimated. It’d be informative to show some efforts of reducing the redundancy in the estimated parameters. Based on Figs. 3-6, it is unclear how the spatial information was incorporated in the classifier (synchronization between EEG channels). It’d be important for this study to show that the high accuracy of SincNet-R was not resulting simply from overfitting (e.g., assigning random labels to the clips and going through the same computation to see if SincNet-R still has the highest accuracy compared with other classifiers). 

Reviewer 2 Report

The paper is very interesting. 

I would like to ask that has authors looked ICA which also solved the cocktail party problem with very good results ?

It would be interesting to reader if add ICA comparison results.

Reviewer 3 Report

The introduction section needs fundamental revision. Some of the sentences are hard to read through. In addition, the background and the clinical applications of this approach were not sufficiently reviewed.

- Line 18-21:

not only in the fields of providing various emotion-based services by creating affective user interface of human-machine interact (HMI) applications, but also in the aspects of helping evaluate psychological diseases for neural disorder patients, such as parkinson’s disorder (PD) [2], autism spectrum disorder (ASD) [3], schizophrenia [4], depression [5], etc.

Please rewrite this sentence.

- Line 29:

electrocardiogram instead of electrocardio 

- Line 15

Section "0.Introduction". Please start from 1 instead of 0.

- Line 63:

but also in its not high classification accuracy due to the subject
dependency of emotion

Please rephrase.

Round 2

Reviewer 1 Report

The revision has taken into account the main comments on the old version.